# Optimizing Treatment of *Staphylococcus aureus* Bloodstream Infections Following Rapid Molecular Diagnostic Testing and an Antimicrobial Stewardship Program Intervention

Hilal Al Sidairi,[a,b] Emma K. Reid,[a] Jason J. LeBlanc,[a,b,c,d] Navjot Sandila,[e] Joline Head,[b] Ian Davis,[a,b,c] Paul Bonnar[a]

[a]Division of Infectious Diseases, Department of Medicine, Dalhousie University, Halifax, Nova Scotia, Canada
[b]Division of Microbiology, Department of Pathology and Laboratory Medicine, Nova Scotia Health, Halifax, Nova Scotia, Canada
[c]Department of Pathology, Dalhousie University, Halifax, Nova Scotia, Canada
[d]Department of Microbiology and Immunology, Dalhousie University, Halifax, Nova Scotia, Canada
[e]Research Methods Unit (RMU), Nova Scotia Health, Halifax, Nova Scotia, Canada

**ABSTRACT** Pending antibiotic susceptibility results, vancomycin is often used for bloodstream infections (BSIs) to ensure treatment of methicillin-resistant *Staphylococcus aureus* (MRSA). As rapid discrimination of methicillin-susceptible *S. aureus* (MSSA) from MRSA in BSIs could decrease vancomycin use and allow early optimization of beta-lactam therapy, this study evaluated the impact of the use of rapid molecular testing for MSSA and MRSA coupled with an antimicrobial stewardship program (ASP) intervention. Between January and July 2020, the Cepheid Xpert MRSA/SA blood culture assay was performed on blood cultures with Gram-positive cocci in clusters that were identified as *S. aureus* using matrix-assisted laser desorption ionization–time of flight mass spectrometry (MALDI-TOF MS). The ASP team member then consulted with the treating physician. The time to optimal therapy (TTOT) and clinical outcomes, including length of hospital stay (LOS), were compared between the intervention ($n = 29$) and historical ($n = 27$) cohorts. TTOT was defined as the time from the first blood culture draw to the use of appropriately dosed antistaphylococcal beta-lactam monotherapy without vancomycin. Molecular testing significantly reduced the median time to MSSA and MRSA discrimination to 7.8 h, compared to 24.3 h with culture-based methods ($P < 0.001$). Compared to the control group, the median TTOT in the ASP intervention group was significantly shorter ($P = 0.041$) at 38.0 h (versus 50.1 h). Rapid discrimination between MRSA and MSSA using molecular testing, paired with an ASP intervention, significantly reduced the TTOT in patients with MSSA BSIs.

**IMPORTANCE** Our research shows that time to optimal antibiotic treatment for serious bloodstream infections can be improved with rapid molecular sensitivity testing and feedback to prescribers. This can be implemented in laboratories without full microbiology services or training to improve patient outcomes by improving antimicrobial use.

**KEYWORDS** stewardship, MRSA, MSSA, *Staphylococcus*, vancomycin, treatment, Xpert, outcome, PCR

**S**taphylococcus aureus bloodstream infections (BSIs) are associated with high morbidity and mortality (1–3). Early and appropriate treatment of *S. aureus* BSIs can reduce mortality, length of hospital stay (LOS), and health care system costs (4–8). The preferred antibiotics for treatment of methicillin-susceptible *S. aureus* (MSSA) BSIs are antistaphylococcal beta-lactams such as nafcillin, oxacillin, and cefazolin (9). However, to ensure coverage for methicillin-resistant *S. aureus* (MRSA), vancomycin is often used empirically, with or without a beta-lactam, prior to the availability of antibiotic susceptibility results from bacterial cultures. If an MSSA infection is confirmed, vancomycin can be discontinued, as beta-lactam monotherapy is associated with improved outcomes (9). With therapy unchanged, vancomycin has an

**Ad Hoc Peer Reviewer** Mary Hopkins

Address correspondence to Paul Bonnar, PaulE.Bonnar@nshealth.ca.

The authors declare no conflict of interest.

**TABLE 1** Laboratory test results for study participants

| Outcome | Data for: | | P value |
| --- | --- | --- | --- |
| | Control group (n = 27) | Intervention group (n = 29) | |
| Time from admission to first blood culture (no. [%]) | | | |
| Proportion <2 days | 6 (22.2) | 7 (24.1) | 0.865 |
| Proportion >2 days | 21 (77.8) | 22 (75.9) | |
| Time from blood culture draw to first positive (median [IQR] [h]) | 21.3 (16.3, 33.1) | 23.0 (19.7, 30.6) | 0.613 |
| Time from first blood culture positive to *S. aureus* MALDI-TOF MS identification (median [IQR] [h]) | | | |
| Using any preparation method | 7.2 (4.3, 9.4) (n = 27) | 5.7 (3.2, 11.1) (n = 29) | 0.353 |
| Using purified blood culture extract | 2.5 (2.3, 2.6) (n = 6) | 3.1 (2.2, 3.8) (n = 13) | 0.195 |
| Using a sample obtained from a plate after 4 to 6 h of growth | 7.2 (7.1, 8.0) (n = 9) | 6.8 (5.9, 7.2) (n = 6) | 0.769 |
| Using an isolated colony from a pure growth culture | 11.0 (7.2, 14.6) (n = 12) | 13.5 (12.1, 19.3) (n = 10) | 0.330 |
| Time from first blood culture positive to any discrimination of MSSA and MRSA (median [IQR] [h]) | 24.3 (21.7, 35.2) | 7.8[a] (4.5, 18.0) | <0.001 |

[a]Based on use of Xpert testing for discrimination of MSSA/MRSA.

unnecessarily broad spectrum of Gram-positive activity and is associated with nephrotoxicity (10). Infectious disease specialists and other members of an antimicrobial stewardship program (ASP) can help guide treatment decisions for the management of BSIs by clinicians to ensure optimal patient outcomes (4, 7, 11–20).

Clinical microbiology laboratories support the diagnosis of BSIs by providing microorganism identification and antimicrobial susceptibilities. Blood culture broths displaying Gram-positive cocci (GPC) in clusters are suggestive of *Staphylococcus* species. With traditional culture methods, differentiation between *S. aureus* and coagulase-negative staphylococci (CoNS) and susceptibility testing to differentiate MSSA from MRSA can take 2 to 3 days (7, 12, 21–23). Over the years, various techniques have been explored to provide more rapid identification of MSSA and MRSA, including using chromogenic media, agglutination reactions, and molecular methods (24). More recently, technologies such as using matrix-assisted laser desorption ionization–time of flight mass spectrometry (MALDI-TOF MS) provide rapid identification of pure microorganisms on solid media, and protocols using MALDI-TOF MS have been established for direct processing of positive blood cultures (13, 16, 22, 23, 25). Despite these successes, antibiotic susceptibilities often lag bacterial identification, and broad-spectrum antibiotics are often used during this period.

In recent years, there has been much interest in rapid phenotypic and molecular methods for determination or deduction of antibiotic susceptibilities, respectively (6, 11, 13, 15–24, 26–32). Due to the wide accessibility of molecular methods, most studies have used these technologies for the rapid detection of *S. aureus* and deduction of MRSA. High sensitivity and specificity for both detection and differentiation of MSSA and MRSA have been shown using the Cepheid Xpert MRSA/SA blood culture assay and other molecular methods (17, 22, 24, 27, 28, 31, 33).

Rapid microorganism identification and susceptibilities, coupled with a well-developed ASP, can ensure optimal patient management and clinical outcomes (11–23, 30–32). This study evaluated the impact of rapid molecular testing for MSSA/MRSA discrimination in *S. aureus* BSIs, followed by an ASP intervention on the time to optimal therapy (TTOT) and clinical outcomes. Given that most blood cultures with GPC in clusters are attributed to CoNS, and with the high cost of molecular testing, this study used an algorithm where the Xpert testing was only performed following the identification of *S. aureus* using MALDI-TOF MS.

## RESULTS

**Patient demographics.** Seventy-eight patients were screened for inclusion. Twenty-two patients were excluded, including 7 in the control group and 15 in the intervention group (see Table S3 in the supplemental material). Demographics (Table S1), laboratory results (Table 1), and primary outcomes (Table 2) are described for the 56 included patients,

**TABLE 2** Comparison of outcomes between control and intervention groups

| Outcome | Data for: | | P value[a] |
| --- | --- | --- | --- |
| | Control group (n = 27) | Intervention group (n = 29) | |
| TTOT (median [IQR] [h]) | 50.1 (29.7, 71.0) | 38.0 (31.5, 53.0) | 0.041 |
| TTAT (median [IQR] [h]) | 46.2 (24.8, 69.5) | 36.4 (27.1, 53.0) | 0.168 |
| LOS (median [95% CI] [days]) | 91.0 (17.0 to NE) | 23.0 (12.0 to 29.0) | 0.024 |
| Duration of vancomycin use (median [IQR] [h])[b] | 14.3 (12.0, 37.3) | 12.0 (12.0, 12.0) | 0.073 |
| Mortality (no. [%])[c] | 7 (26) | 3 (11) | |
| Readmission (no. [%])[c] | 4 (15) | 1 (4) | |
| Bacteremia relapse (no. [%])[c] | 3 (12) | 0 | |

[a]P value for equality of group Kaplan-Meier curves.
[b]Patients who received more than 1 dose of vancomycin only occurred in a subset of participants (n = 6 and n = 11 for the intervention and control groups, respectively).
[c]At day 30 post completion of definitive antibiotic therapy.

29 in the intervention and 27 in the control groups. No significant differences were noted for age; gender; medical conditions, including diabetes or chronic renal disease; or admitting medical service (Table S1).

**Laboratory results.** The median time from blood culture positivity to identification of *S. aureus* varied by MALDI-TOF MS preparation method (Table 1). In the historical cohort, there were 6 specimens processed for MALDI-TOF MS using purified blood culture extract, 9 using a sample obtained from a plate after 4 to 6 h of growth, and 12 using an isolated colony from a pure growth culture, whereas in the intervention group, there were 13, 6, and 10, respectively. Despite more isolates identified using the quicker purified blood culture extract in the intervention group, there were no significant differences observed in time to *S. aureus* identification between the control and intervention groups overall or among the various MALDI-TOF MS preparation methods (Table 1). Regardless of the processing method used to identify *S. aureus* with MALDI-TOF MS, the median time from positive blood culture to final identification and susceptibility result was significantly lower ($P < 0.001$) in the intervention group at 7.8 h than in the control group at 24.3 h (Table 1).

**Clinical outcomes.** The median TTOT was significantly ($P = 0.041$) shorter in the intervention group at 38.0 h than in the historical cohort at 50.1 h (Table 2). At any time during the study period, participants in the intervention group were 77% more likely ($P = 0.043$) to start optimal therapy than participants in the control group, with a hazard ratio (HR) of 1.77 (95% confidence interval [CI], 1.02 to 3.09) (Table S4). While there was a shorter time to appropriate therapy (TTAT) and less vancomycin use in the intervention group, these differences were not significant (Table 2). In the control group, 11 patients (41%) were prescribed more than 1 dose of vancomycin compared to 6 (21%) in the intervention group. The median LOS was found to be significantly shorter in the intervention group at 23 days than in the control group at 91 days ($P = 0.024$) (Table 2). Of the 56 patients, 9 had hospital stays of 60 days or longer. Two of these prolonged hospital stays took place in the intervention group (80 and 91 days), and the remaining 7 were in the control group (ranging from 91 to 360 days). The majority (6/9) of prolonged stays, however, were due to deconditioning and functional decline in hospital that required rehabilitation and/or lengthy waits while transitioning into long-term care. Within the intervention group, the rate of ASP recommendation acceptance for antibiotic choice within 24 h was 100%. Formal infectious disease consults occurred in 86.2% (25/29) of intervention group participants and 66.7% (18/27) of control participants. At day 30 post completion of definitive antibiotic therapy, the odds of mortality for the intervention group was 0.36 (95% CI, 0.08 to 1.56; $P = 0.17$), the odds of readmission was 0.21 (95% CI, 0.02 to 2.04; $P = 0.18$), and the odds of bacteremia relapse was 0.12 (95% CI, 0.01 to 2.63; $P = 0.18$) compared to the control group (Table 2).

## DISCUSSION

In this study, the implementation of rapid microbiological identification and susceptibilities for *S. aureus* BSI coupled with an ASP intervention significantly reduced the time to susceptibility results and TTOT, which ensured de-escalation to appropriately dosed antistaphylococcal beta-lactam monotherapy. The TTAT and vancomycin use were lower in the intervention

group but did not reach statistical significance. The TTAT reduction may not have been significant due to a shorter TTAT than TOTT (46.2 h versus 50.1 h, respectively) in the control group, suggesting empirical vancomycin started before beta-lactam therapy. Therefore, the improvement during the intervention was less dramatic for TTAT than TTOT. Though vancomycin therapy with or without a beta-lactam will sufficiently treat *S. aureus* BSI, monotherapy with an antistaphylococcal beta-lactam was deemed optimal since vancomycin is associated with nephrotoxicity, is less effective than beta-lactam antibiotics for MSSA BSI, and has broader-spectrum Gram-positive activity (9). Appropriate antibiotic therapy can improve BSI outcomes and reduce mortality (10), as well as reduce LOS in hospitals (5) and associated health care costs (6–8).

In recent years, many studies have explored rapid microbiological identification and susceptibilities, with goals to improve the TTOT of BSIs (6, 13, 15–17, 19, 21–23, 27, 31, 32, 34). For example, Spencer et al. (27) demonstrated that molecular testing can be used on blood cultures showing GPC in clusters and reduce the time to MSSA and MRSA detection from a range of 30 to 50 h with conventional culture and susceptibility testing to just 1 to 5 h with rapid molecular testing. Despite this, Frye et al. (29) demonstrated that the benefits of molecular testing are only realized when the test is available on-demand and results are followed up with interventions. Unfortunately, on-demand testing does not occur in laboratories unless 24/7 operations are available, along with a technologist trained in the required methods.

Our study demonstrated the benefits of rapid molecular testing following identification of *S. aureus* from MALDI-TOF MS. There was variation in the methods used to prepare specimens for MALDI-TOF MS identification when comparing the historical cohort and the intervention group (Table 1). Thirteen isolates were identified using the purified blood culture extract method in the intervention group compared to 6 for the control group. However, this likely did not influence outcomes since the overall time to MALDI-TOF MS identification was the same in both groups and MALDI-TOF MS did not provide any information on sensitivities. As seen in Table 1, further benefits could be achieved if more specimens were performed using purified blood culture extract for MALDI-TOF MS and Xpert testing, as only 44.8% (13/29) of specimens in the intervention group benefited from this optimal testing algorithm.

Xpert testing significantly reduced the time for overall MSSA and MRSA discrimination compared to culture-based methods, from 24.3 to 7.8 h. When paired with an ASP intervention, rapid molecular testing decreased the median TTOT, measured from the time of the first blood culture draw, to 38 h compared to 50 h in the historical cohort, which is consistent with others using similar methods and considerations (7, 11–13, 16, 19, 21). When accounting for the time required for blood culture incubation (median time from blood culture draw to the first positive of 23 h in the intervention group), the time to optimal therapy from first blood culture positivity would be 15 h. Since it took 7.8 h for MSSA and MRSA discrimination using Xpert, there was a gap of about 7 h from the Xpert result to optimal therapy. This is not unexpected given delays in the ASP feedback process, including contacting the appropriate physician, time of day, and delays in writing the order.

Other studies and methods have demonstrated that rapid microbiological identification and susceptibilities can improve the TTOT, and the benefits are most effective when paired with an ASP intervention (6, 13, 15–17, 19, 21–23, 27, 31, 32, 34). However, few studies have been able to clearly demonstrate benefits to clinical outcomes of decreased TTOT, likely due to the large number of study participants required to achieve these study endpoints (12, 21). Compared to traditional methods, Page et al. (21) showed a reduction from 55.5 h to 43.5 h of antibiotic use with implementations of Xpert testing on blood cultures from obstetric patients and a significant reduction in LOS from 65.5 h to 56 h. With Xpert testing and infectious diseases pharmacist notification, Bauer et al. (6) observed a quicker switch from empirical vancomycin to antistaphylococcal beta-lactams by 1.7 days, a decrease in LOS of 6.2 days, and fewer associated health care costs. In our study, there was a significant reduction in median hospital LOS in the intervention group at 23 days compared to 91 days in the control group. Given the small number of study participants, it is likely that confounders are responsible for most of this difference. In our study, some patients were in the hospital for a prolonged period and likely skewed our LOS results. Nine had hospital stays of 60 days or longer, and 7 such patients

were in the control group (ranging from 91 to 360 days). These long hospital stays in our study were related to deconditioning and functional decline after initial infection treatment, which occurred more often in the control group. ASP interventions might have contributed to reducing the LOS with guidance on *S. aureus* management, such as the need to document BSI clearance, duration of antibiotic treatment, the need to perform echocardiography, and recommendation for infectious disease consultation. While such interventions may delay discharge in a few patients to pursue this workup, early and optimal management of MSSA BSI likely reduces delayed recognition of complications which often become more severe over time, such as septic joints, vertebral osteomyelitis, and abscess. Furthermore, infectious diseases consultation does improve outcomes (3), and this occurred more often in our intervention group than in the control participants (86.2% and 66.7%, respectively).

The methods used in this study are applicable and scalable in any laboratory and hospital using these technologies with access to a well-developed ASP. In hospitals without a formal ASP, a standardized form (see Table S2 in the supplemental material) for guidance on MSSA BSI management could be developed (by ASP team members, infectious disease specialists, and pharmacists) for laboratory technologists or microbiologists, who could provide preliminary feedback to the prescribing physicians following the rapid testing results. This approach could be of value for any hospital to reduce variability and strengthen acceptability of ASP recommendations. In rural areas without access to complete microbiology services, blood cultures or the workup of positive blood cultures may be performed in outside laboratories, which significantly delays final reports. As such, the potential benefits of rapid diagnostics would likely be more profound than observed in this study. It should be noted that Xpert testing is among the simplest of molecular methods, requiring only a few manual steps for processing and obtained results from the manufacturer's software. On the other hand, without access to MALDI-TOF MS for rapid discrimination of *S. aureus* from CoNS, other technologies may be needed, such as point-of-care (POC) testing for *S. aureus* or MRSA (30). Yossepowitch et al. (30) used a rapid immunochromatographic test for *S. aureus* BSI following the identification of GPC in clusters from positive blood cultures and demonstrated a sensitivity and specificity of 92% and 99%, respectively, all while reducing cost by 75%. In laboratories without a technologist to perform and interpret a Gram stain to identify GPC in clusters, the benefits of rapid testing should be considered against the cost of testing all positive blood cultures. Testing all positive blood cultures with Xpert would have no impact on nonstaphylococcal microorganisms, but there would also be benefits for early identification of MRSA and CoNS. Davies et al. (34) used Xpert testing and demonstrated de-escalation of vancomycin to antistaphylococcal beta-lactams in 25% of MSSA BSIs, as well as earlier treatment of MRSA BSIs with vancomycin in 54% of BSIs and antibiotic discontinuation when MSSA or MRSA was absent. The impact of staphylococcal testing strategies in other settings may differ depending on local context, including MRSA prevalence, infectious disease availability, blood culture contamination rates, overall positivity rates, cost, and ASP availability.

Altogether, physicians require rapid identification and antibiotic susceptibility testing to make informed decisions on targeted antibiotic therapy. In a population with a prevalence of 10% to 12% MRSA, this study demonstrated the benefits on TTOT of rapid MSSA and MRSA detection coupled with ASP intervention during routine work hours. Further studies will explore the incremental benefits of 24/7 microbiology and ASP services and testing strategy amenable for optimal benefits in rural areas.

## MATERIALS AND METHODS

**Study design and patient selection.** Prospective blood cultures collected from January to July 2020 from inpatients in a tertiary care hospital were compared to a historical cohort of samples from patients with similar demographics from December 2017 to March 2018 (see Table S1 in the supplemental material). Specimens from the historical cohort were processed using identical methods apart from molecular testing and ASP intervention. Study inclusion criteria were patients aged ≥18 years with a monomicrobial MSSA BSI. For patients with concurrent or recurrent positive blood cultures, only the first with *S. aureus* was considered for the study. Exclusion criteria included prior colonization with MRSA, polymicrobial BSIs, transfer from another hospital with *S. aureus* BSI, antibiotic allergy (cloxacillin, cefazolin, or vancomycin), or if patients were transitioned to comfort care. In Canada, cloxacillin and cefazolin are the antistaphylococcal beta-lactams available.

Patients who received definitive therapy with beta-lactams that were not cloxacillin or cefazolin (e.g., piperacillin-tazobactam, or meropenem) were excluded from outcome analysis since such patients likely had confounding characteristics not representative of patients with monomicrobial *S. aureus* BSI. Patients were excluded from the intervention group if samples were identified by MALDI-TOF MS when ASP was not available, primarily outside routine working hours (0800 to 1600, Monday to Friday).

**Standard laboratory methods.** Blood cultures were performed using Bactec Plus Aerobic/F or Bactec Lytic/10 Anaerobic/F culture vials (Becton, Dickinson and Company, Sparks, USA) and were incubated onto the Bactec FX instrument as soon as they were received in the laboratory. For each patient, the first blood culture vial flagging positive by the instrument was subjected to Gram staining and inoculated onto BD BBL Trypticase soy agar with 5% sheep blood agar (TSA II), BD BBL CDC anaerobic blood agar, and enriched chocolate agar (Oxoid, Napean, Canada). Blood cultures showing GPC in clusters were subjected to MALDI-TOF MS on a Vitek MS (bioMérieux, Marcy-Étoile, France) as per manufacturer recommendations. As per local protocols, depending on the time of day and availability of trained technologists, MALDI-TOF MS was performed directly on an aliquot from blood culture vials from a sample obtained after 4 to 6 h of growth or from isolated colonies after 8 to 12 h of growth.

Direct processing of blood for MALDI-TOF MS was performed as previously described (25). Briefly, 1 mL of blood was lysed with 200 $\mu$L of 5% saponin (Sigma-Aldrich, St. Louis, USA) and vortexing. After a 5-min incubation, the lysate was centrifuged for 2 min at 13,000 $\times$ *g*, and the supernatant was discarded. The pellet was washed with 1 mL sterile distilled water and resuspended in 100 $\mu$L of water and 300 $\mu$L 100% ethanol (Fisher Scientific, Pittsburgh, USA). Following centrifugation, the supernatant was carefully removed and mixed with 50 $\mu$L 98% formic acid (Merck kGaA, Billerica, MA, USA) and 50 $\mu$L of 99.5% acetonitrile (Fisher Scientific).

Either 1 $\mu$L of purified blood culture extract, a sample obtained from a plate after 4 to 6 h of growth, or an isolated colony from a pure growth culture was covered in Vitek MS $\alpha$-cyano-4-hydroxycinnamic acid matrix and subjected to MALDI-TOF MS analysis using a Vitek MS *in vitro* diagnostic (IVD) system (bioMérieux, Marcy-Étoile, France). Spectra generating identifications with >98% confidence were accepted at the species level, and blood cultures with *S. aureus* were subjected to molecular testing.

**Phenotypic and molecular-based antibiotic susceptibility testing.** Antibiotic susceptibilities were performed on all *S. aureus* cultures using an AST-GP67 card (bioMérieux, Durham, NC, USA) on the Vitek 2 instrument as per manufacturer recommendations. In the intervention group, isolates identified as *S. aureus* by MALDI-TOF were then characterized as MSSA or MRSA using the Xpert MRSA/SA BC assay (Cepheid, Sunnyvale, CA, USA) as per manufacturer recommendations. Briefly, 50 $\mu$L of the positive blood culture was added to the elution reagent vial provided by the manufacturer, and after mixing, the entire contents were transferred to the sample reagent chamber of the Xpert cartridge. The cartridge was loaded on a GeneXpert Dx instrument and interpreted by the manufacturer's software.

**Antimicrobial stewardship intervention.** Once *S. aureus* was identified from a blood culture, a microbiology laboratory technologist contacted an ASP physician or clinical pharmacist during routine working hours (0800 to 1600, Monday to Friday). The ASP team member verified molecular or phenotypic susceptibilities and contacted the treating physician with the results by telephone. Xpert results were only released if an ASP team member was available to review and phone the treating team. Standardized guidance provided included discussion on appropriate narrowing to optimal antimicrobial therapy, dosage adjustments, timing of repeat blood cultures, echocardiography, and referral for an infectious diseases physician consult (Table S2).

**Data sources and ethics.** This study was approved by the Nova Scotia Health Research Ethics Board under file number 1025184. Data were collected from electronic patient records using the Research Electronic Data Capture software and included patient demographics (Table S1), laboratory results (Table 1), and outcome data related to inpatient antibiotics therapy and endpoints at 30 days from completion of definitive antimicrobial therapy (Table 2).

**Clinical outcomes.** The primary outcome of TTOT was defined as the time (in hours) from the first blood culture draw to starting optimal single-agent therapy for MSSA (i.e., time of first appropriately dosed cefazolin or cloxacillin monotherapy). Secondary outcomes included the time to appropriate therapy (TTAT), defined as the time from the first blood culture draw to the time of starting appropriate therapy for MSSA with or without vancomycin. Additional data of interest included total duration of vancomycin use for patients who were prescribed more than 1 dose, LOS, and acceptance of ASP recommendations for antibiotic therapy. At day 30 postcompletion of definitive antibiotic therapy, the rates of all-cause mortality, hospital readmission, and bacteremia relapse were collected.

**Statistical analyses.** A sample size of 31 in each group was required to detect a reduction in mean time of reporting *S. aureus* susceptibility results from 30 h to 20 h, with statistical power of 91% and two-sided significance level of 0.05. Baseline demographics and laboratory test results were summarized as mean (standard deviation), median (interquartile range [IQR]), or frequency (%). Differences between groups were compared using Student's *t* test or Wilcoxon rank-sum test for continuous variables and Pearson's chi-square tests or Fisher's exact test for categorical variables. TTOT, TTAT, and duration of vancomycin use were characterized using Kaplan-Meier plots, and the log-rank test was used to compare outcomes. LOS was characterized using a competing risk model, with discharge and death considered competing risks, and Gray's test was used for comparisons. Cox proportional-hazards models were used to estimate the unadjusted hazard ratios (HRs), and the Kolmogorov-type supremum test was used to assess the proportional hazards assumption. Univariate logistic regression analyses were performed for death at 30 days, readmission at 30 days, and bacteremia relapse at 30 days. Due to quasicomplete separation in relapse at 30 days, the Firth logistic regression method was implemented. A two-sided *P* value of <0.05 was the threshold for statistical significance. All analyses were performed using SAS statistical software version 9.4 (SAS Institute Inc., Cary, USA).

## SUPPLEMENTAL MATERIAL

Supplemental material is available online only.

**SUPPLEMENTAL FILE 1**, PDF file, 0.2 MB.

## ACKNOWLEDGMENTS

We thank the Nova Scotia Health Research Fund, the Research Methods Unit, and the in-kind contributions of the Division of Microbiology in the Central Zone Nova Scotia Health for microbiology services, including Xpert testing. No industry sponsors were involved in the study funding, design, or result interpretation.

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
