## [Reviewer comments · Microbiology Spectrum]

Microbiology Spectrum

Optimizing treatment of *Staphylococcus aureus* bloodstream infections following rapid molecular diagnostic testing and an antimicrobial stewardship program intervention

Hilal Al Sidairi, Emma Reid, Jason LeBlanc, Navjot Sandila, Joline Head, Ian Davis, and Paul Bonnar

Corresponding Author(s): Paul Bonnar, Nova Scotia Health Authority

Review Timeline:

Submission Date:	May 4, 2022
Editorial Decision:	June 11, 2022
Revision Received:	August 9, 2022
Editorial Decision:	August 19, 2022
Revision Received:	January 19, 2023
Accepted:	January 26, 2023

Editor: Tulip Jhaveri

Reviewer(s): Disclosure of reviewer identity is with reference to reviewer comments included in decision letter(s). The following individuals involved in review of your submission have agreed to reveal their identity: Mary Hopkins (Reviewer #2)

Transaction Report:

DOI: <https://doi.org/10.1128/spectrum.01648-22>

June 11, 2022

Dr. Paul Bonnar
Nova Scotia Health
5078 Dickson Bldg, 5780 University Ave
Halifax, Nova Scotia B3H2Y1
Canada

Re: Spectrum01648-22 (Optimizing treatment of *Staphylococcus aureus* bloodstream infections following rapid molecular diagnostic testing and an antimicrobial stewardship program intervention)

Dear Dr. Paul Bonnar:

Link Not Available

Sincerely,

Tulip Jhaveri

Journals Department
Reviewer comments:

Reviewer #1 (Comments for the Author):

This paper describes a reduction in time to optimal therapy after use of PCR-based method for rapid identification of MSSA bloodstream infections.

Please review the article and ensure formatting is standard throughout. For one example, hours is denoted as h or hours at various points and should be uniform throughout the manuscript. It should be Gram-positive, not Gram positive. Please also ensure all abbreviations defined in initial abbreviation section, PCR, CHCA, and hazard ratio all defined in text and not included here (may also be others). p value is also capitalized throughout manuscript. Spacing is varied surrounding = and < and should

be uniform.

If required sample size was 31 to power the study, why were 31 patients not used in the results section? Would recommend increasing sample size to be 31 patients in intervention and control arms given statistical analysis stating a need for this size

Did the providers have access to the Xpert results and if able to interpret independently deescalate antibiotics without the use of the ASP team on evenings/weekends? If so, this would also decrease the TTOT in the intervention arm

Line 63: would include methicillin/oxacillin as cloxacillin is not a conventional treatment for MSSA BSI in US

Line 68: only single toxicity described (kidney), change sentence to reflect this or add additional toxicity examples such as adverse reactions

Line 102: monomicrobial is a single word not hyphenated

Line 120: please define what a scum plate is as plating on this is not described in the previous section describing inoculation of first positive blood cultures on various plates and the terminology "scum plate" is not common knowledge. Please use timeframe rather than "overnight growth" since timeframe is used for other descriptions and describe if this is from one of the plates or a liquid culture

Line 152: TTOT is already defined in prior section (line 41). Please clarify how TTOT and TTAT are different as this sentence is very hard to comprehend. For example, TTOT is defined as the time from first blood culture draw to optimal single-agent therapy for MSSA (i.e. time of first appropriately dosed cefazolin or cloxacillin monotherapy or time at which vancomycin was discontinued) and TTAT is defined as the time from the first blood culture draw to starting appropriate therapy for MSSA with or without vancomycin.

Line 163: *S aureus* should be italicized

Line 196: add that 50.1h is for the control group

Line 198: define CI is confidence interval

Line 199: what does continuous vancomycin use mean? Is this different from total vancomycin use? If TTOT is shorter in intervention arm would expect total vancomycin use be less in the intervention arm as well so please explain this result if not true

Line 204: please relist the 30 day outcomes that were studied

Line 211: relates back to comments about line 199, was vancomycin discontinued earlier in the intervention arm?

Line 214: would rephrase this as excessive is not the correct terminology, consider focusing on improved outcomes for patients treated with anti-Staph b-lactam rather than spectrum of activity of vancomycin, or can write broader spectrum Gram-positive activity rather than excessive

Line 219: I find this sentence confusing, please extend sentence to state what describing ideal testing conditions means and what "potential benefits < 5 hours from blood culture positivity" means

Line 229: please edit: there was no difference in the time from blood culture draw to first positive result in the intervention or control arm

Line 232: did the hours technologists were working or the number of experienced technologists able to perform testing change in the time of the control versus intervention? Would explain more about this if this is felt to be why the time to MALDI-ToF identification was felt to be due to this

Line 251: Please rephrase the sentence about the Bauer paper so it is more clear

Line 253: The way this sentence is written it sounds like it is referring to the Bauer paper, would rephrase and include a transition sentence

Line 259: all of reasons described sound like things that would increase length of stay (addition testing, consultation), would look at patient charts to help hypothesize what confounders might be or an explanation for shorter LOS in intervention

Line 278: What does "testing all blood cultures with Xpert would have no impact for non-staphylococcal microorganisms, but the benefits would not only include those of MSSA BSIs" mean? Should it read the benefits WOULD only include those of MSSA

BSIs?

Line 283: please rephrase this final sentence, it does not make sense

Table 2: as in results section, do not fully understand what continuous vancomycin use means

Reviewer #2 (Comments for the Author):

As the authors mentioned the vast difference in LOS between the intervention arm and the control for this relatively small sized study likely has very little to do with the modest improvement in TTOT. Consider expounding a little more on this >2 month discrepancy in median LOS.

Staff Comments:

Preparing Revision Guidelines

Please return the manuscript within 60 days; if you cannot complete the modification within this time period, please contact me. If you do not wish to modify the manuscript and prefer to submit it to another journal, please notify me of your decision immediately so that the manuscript may be formally withdrawn from consideration by Microbiology Spectrum.

We have added point-by-point responses to the issues raised by the reviewers below.

Reviewer #1 (Comments for the Author):

This paper describes a reduction in time to optimal therapy after use of PCR-based method for rapid identification of MSSA bloodstream infections.

Please review the article and ensure formatting is standard throughout. For one example, hours is denoted as h or hours at various points and should be uniform throughout the manuscript. It should be Gram-positive, not Gram positive. Please also ensure all abbreviations defined in initial abbreviation section, PCR, CHCA, and hazard ratio all defined in text and not included here (may also be others). p value is also capitalized throughout manuscript. Spacing is varied surrounding = and < and should be uniform.

- Standardized hours to “h”
- Other articles in Microbiology Spectrum use *P* (italicized and capitalized) so formatted to match
- Gram positive changed to Gram-positive
- Removed “PCR”. Was only sporadically used in paper and always after “Xpert”. For conciseness just using “Xpert”
- Removed CHCA since only used the 1 time
- Updated/standardized the industry location formatting
- Added interquartile range (IQR), hazard ratios (HR)
- Spacing standardized surrounding = and <.

If required sample size was 31 to power the study, why were 31 patients not used in the results section? Would recommend increasing sample size to be 31 patients in intervention and control arms given statistical analysis stating a need for this size

- We initially screened 78 patients but only had data available for 56 patients (29 in the intervention and 27 in the control groups). The other patients not included since there was data missing on chart review. The power calculation was based on an estimated reduction in sensitivity reporting by 10 hours. In our study, the time for overall MSSA/MRSA discrimination compared to, was reduced from 24.3 h using culture-based methods to 7.8 h with Xpert, a difference of 16.5). In addition, we were able to achieve statistical significance for our primary outcome, so more patients were not added to the study.

Did the providers have access to the Xpert results and if able to interpret independently deescalate antibiotics without the use of the ASP team on evenings/weekends? If so, this would also decrease the TTOT in the intervention arm

- Results were only released on a laboratory report if ASP was called and verified the Xpert test. Therefore, no reporting of Xpert results were released without ASP feedback. Added this clarification to the manuscript.

Line 63: would include methicillin/oxacillin as cloxacillin is not a conventional treatment for MSSA BSI in US

- Changed to “...such as nafcillin, oxacillin, or cefazolin.”
- Added following in methods section: In Canada, cloxacillin and cefazolin are the anti-staphylococcal beta-lactams available.

Line 68: only single toxicity described (kidney), change sentence to reflect this or add additional toxicity examples such as adverse reactions

- Done, changed to nephrotoxicity

Line 102: monomicrobial is a single word not hyphenated

- Done

Line 120: please define what a scum plate is as plating on this is not described in the previous section describing inoculation of first positive blood cultures on various plates and the terminology "scum plate" is not common knowledge. Please use timeframe rather than "overnight growth" since timeframe is used for other descriptions and describe if this is from one of the plates or a liquid culture

- Changed "scum" to 4-6 hour growth.
- Overnight: 8-12 hours

Line 152: TTOT is already defined in prior section (line 41). Please clarify how TTOT and TTAT are different as this sentence is very hard to comprehend. For example, TTOT is defined as the time from first blood culture draw to optimal single-agent therapy for MSSA (i.e. time of first appropriately dosed cefazolin or cloxacillin monotherapy or time at which vancomycin was discontinued) and TTAT is defined as the time from the first blood culture draw to starting appropriate therapy for MSSA with or without vancomycin.

- The first definition (line 41) is in abstract, the second is in body of paper so required repeating the definition.
- Clarified wording.

Line 163: *S aureus* should be italicized

- done

Line 196: add that 50.1h is for the control group

- Reworded

Line 198: define CI is confidence interval

- done

Line 199: what does continuous vancomycin use mean? Is this different from total vancomycin use? If TTOT is shorter in intervention arm would expect total vancomycin use be less in the intervention arm as well so please explain this result if not true

- Continuous vancomycin use was when prescribed for more than 1 dose. Agree confusing term, so changed to just vancomycin duration and added the note that only included patients who received more than 1 dose.
- Reworded, there was less vancomycin use but not significantly so
- Added "The TTAT and vancomycin use was lower in the intervention group but did not reach statistical significance" to start of discussion.

Line 204: please relist the 30 day outcomes that were studied

- Reworded, since we listed in the subsequent sentence

Line 211: relates back to comments about line 199, was vancomycin discontinued earlier in the intervention arm?

- Added The TTAT and vancomycin use was lower in the intervention group but did not reach statistical significance. Added more explanation.

Line 214: would rephrase this as excessive is not the correct terminology, consider focusing on improved outcomes for patients treated with anti-Staph b-lactam rather than spectrum of activity of vancomycin, or can write broader spectrum Gram-positive activity rather than excessive

- changed

Line 219: I find this sentence confusing, please extend sentence to state what describing ideal testing conditions means and what "potential benefits < 5 hours from blood culture positivity" means

- Removed sentence as confusing and adding any information

Line 229: please edit: there was no difference in the time from blood culture draw to first positive result in the intervention or control arm

- done

Line 232: did the hours technologists were working or the number of experienced technologists able to perform testing change in the time of the control versus intervention? Would explain more about this if this is felt to be why the time to MALDI-ToF identification was felt to be due to this

- We are unable to dissect out but likely just real-world variability since there were no major lab practice changes or changes in training standards amongst the technologists.

Line 251: Please rephrase the sentence about the Bauer paper so it is more clear

- done

Line 253: The way this sentence is written it sounds like it is referring to the Bauer paper, would rephrase and include a transition sentence

- Removed "...did not show a reduction in 30-day outcomes (i.e., mortality, re-admission, or bacteremia relapse)..."
- added transition

Line 259: all of reasons described sound like things that would increase length of stay (addition testing, consultation), would look at patient charts to help hypothesize what confounders might be or an explanation for shorter LOS in intervention

- Added clarifying text

Line 278: What does "testing all blood cultures with Xpert would have no impact for non-staphylococcal microorganisms, but the benefits would not only include those of MSSA BSIs" mean? Should it read the benefits WOULD only include those of MSSA BSIs?

-Change to “Testing all positive blood cultures with Xpert would have no impact for non-staphylococcal microorganisms, but there would also be benefits for early identification of MRSA and CONS.”

Line 283: please rephrase this final sentence, it does not make sense

- reworded

Table 2: as in results section, do not fully understand what continuous vancomycin use means

- Modified wording

Reviewer #2 (Comments for the Author):

As the authors mentioned the vast difference in LOS between the intervention arm and the control for this relatively small sized study likely has very little to do with the modest improvement in TTOT.

Consider expounding a little more on this >2 month discrepancy in median LOS.

- Added more reasoning including example of patients who may be awaiting long-term care placement skewing results.

- Also softened the statement in the abstract conclusion.

August 19, 2022

Dr. Paul Bonnar
Nova Scotia Health
5078 Dickson Bldg, 5780 University Ave
Halifax, Nova Scotia B3H2Y1
Canada

Re: Spectrum01648-22R1 (Optimizing treatment of *Staphylococcus aureus* bloodstream infections following rapid molecular diagnostic testing and an antimicrobial stewardship program intervention)

Dear Dr. Paul Bonnar:

Thank you for submitting your manuscript to Microbiology Spectrum. Minor revisions still needed before I can consider accepting the manuscript. When submitting the revised version of your paper, please provide (1) point-by-point responses to the issues raised by the reviewers as file type "Response to Reviewers," not in your cover letter, and (2) a PDF file that indicates the changes from the original submission (by highlighting or underlining the changes) as file type "Marked Up Manuscript - For Review Only". Please use this link to submit your revised manuscript - we strongly recommend that you submit your paper within the next 60 days or reach out to me. Detailed instructions on submitting your revised paper are below.

Link Not Available

Sincerely,

Tulip Jhaveri

Journals Department
Reviewer comments:

Reviewer #1 (Comments for the Author):

This paper describes rapid molecular testing in concert with antimicrobial stewardship intervention on the time to optimal therapy for MSSA bacteremia. The authors found that use of the testing with a direct intervention by their stewardship team decreased the time discrimination of MSSA vs MRSA and the time to optimal therapy. This finding is not novel but still important. One of the biggest weaknesses in the paper is the 3 different means of identifying organisms by MALDI which changes the time to ID depending on which method was used. If the same method was used for all intervention and control samples the paper would be much stronger.

One thing that I would recommend is going back to understand why the length of stay was so much longer in the control group. It should be possible to open charts and review given the small numbers in the study to understand this or if there are particular outliers which are skewing the average as you would not expect the difference to be so stark.

Line 87: please describe the testing methods for the Cepheid Xpert

Line 127: Is there any difference in success technique by MALDI depending on the preparation method (direct from blood culture vs sample after 4-6 hours of growth vs colonies after 8-12 hours of growth)?

Line 179: please describe why demographic, lab value, and primary outcome data is not available for 22 patients, this is a significant % of study population and makes it very hard to know if there are confounders present

Line 185: please describe how many isolates in each arm were identified by each route on the MALDI if the delay in time to ID in the intervention arm was felt to be due to lack of trained technologists to run the MALDI directly from blood cultures

Line 201: are there particularly characteristics of teams or patients who did not accept the AST recommendations within 24 hours?

Line 230: Given intervention was only on the weekdays were any patients with positive blood cultures on the weekends automatically excluded from the trial?

Line 259: as with earlier comment, please look through the chart and describe what confounders are responsible for a difference in hospital length of stay

Line 289: typo for CoNS

Table 2: please include mortality in each group

Supplemental table 2: font size and type is not uniform in this table

Staff Comments:

Preparing Revision Guidelines

Please return the manuscript within 60 days; if you cannot complete the modification within this time period, please contact me. If you do not wish to modify the manuscript and prefer to submit it to another journal, please notify me of your decision immediately so that the manuscript may be formally withdrawn from consideration by Microbiology Spectrum.

Reviewer comments:

Reviewer #1 (Comments for the Author):

This paper describes rapid molecular testing in concert with antimicrobial stewardship intervention on the time to optimal therapy for MSSA bacteremia. The authors found that use of the testing with a direct intervention by their stewardship team decreased the time discrimination of MSSA vs MRSA and the time to optimal therapy. This finding is not novel but still important. One of the biggest weaknesses in the paper is the 3 different means of identifying organisms by MALDI which changes the time to ID depending on which method was used. If the same method was used for all intervention and control samples the paper would be much stronger.

One thing that I would recommend is going back to understand why the length of stay was so much longer in the control group. It should be possible to open charts and review given the small numbers in the study to understand this or if there are particular outliers which are skewing the average as you would not expect the difference to be so stark.

- Agree, the LOS of skewed mainly due to a few patients that had prolonged stay in hospital. Updated results and discussion to reflect this.
- Out of 56 patients, 9 had hospital stays of 60 days or longer. Two of these prolonged hospital stays took place in the intervention group (80 and 91 days), and the remaining seven were in the control group (ranging from 91 to 360 days). Two of nine required ICU admission, both in the control group. Three patients had complicated infections including infective endocarditis with aortic valve insufficiency, a prosthetic joint infection, and CNS infection with concurrent osteomyelitis, which would each in general require longer antibiotic courses and would contribute to longer hospital stays. The majority (6 out of 9) of prolonged stays, however, can conceivably be explained by deconditioning and functional decline in hospital that required rehabilitation and/or lengthy waits while transitioning into long-term care.
- Given these issues, removed following from abstract: “and the median LOS was significantly shorter ($P=0.024$) at 23 days (vs. 91 days).”

Line 87: please describe the testing methods for the Cepheid Xpert

- Added following to **Phenotypic and molecular-based antibiotic susceptibility testing** results section: “In the intervention group, *S. aureus* strains identified by MALDI-ToF were characterized as MSSA and MRSA using the Xpert MRSA/SA BC assay (Cepheid, Sunnyvale, USA), as per manufacturer recommendations. Briefly, 50 μ l of the positive blood culture was added to the Elution Reagent vial provided by the manufacturer, and

after mixing, the entire contents were transferred to the sample reagent chamber of the Xpert cartridge. The cartridge was loaded on a GeneXpert Dx instrument and interpreted by the manufacturer software.”

Line 127: Is there any difference in success technique by MALDI depending on the preparation method (direct from blood culture vs sample after 4-6 hours of growth vs colonies after 8-12 hours of growth)?

- See response to Line 185 below. No MALDI preparation methods failed to identify *S. aureus* during the study period.

Line 179: please describe why demographic, lab value, and primary outcome data is not available for 22 patients, this is a significant % of study population and makes it very hard to know if there are confounders present

- Clarified the wording in manuscript. Only 56 patients were included in the study. Therefore, 22 patients were excluded. The reasons for exclusion were added in new Table S3

Line 185: please describe how many isolates in each arm were identified by each route on the MALDI if the delay in time to ID in the intervention arm was felt to be due to lack of trained technologists to run the MALDI directly from blood cultures

- We appreciate the opportunity to provide more details into the contribution of each MALDI preparation methods for each arm. Additional data has been added to Table 1 to accommodate this suggestion, as well as a more thorough explanation in the results section. Since we now have how many isolates in each arm were identified by each route on the MALDI, we have removed the prior “Time (h) from first blood culture positive to *S. aureus* identification; median (IQR)” variable in Table 1 and the text. In the historical cohort, there were 6, 9, and 12 specimens processed for MALDI-ToF using direct blood, a scum plate, or a cultured isolate, whereas in the intervention group, there were 13, 6, and 10, respectively. Despite subtle differences in numbers processed by each method, there were no significant differences observed between MALDI-ToF preparation methods (Table 1).
- Discussion also updated

Line 201: are there particularly characteristics of teams or patients who did not accept the AST recommendations within 24 hours?

- When we re-evaluated, there were two coding errors so actually 100% of pts has AMS advice re antimicrobials at 24 hours accepted. The miscoding was in two patients that were already on cefazolin correctly – so in fairness they should have been excluded and not coded as failure to accept AMS advice.

Line 230: Given intervention was only on the weekdays were any patients with positive blood cultures on the weekends automatically excluded from the trial?

- Correct, in methods under **Antimicrobial stewardship intervention**, we specified: Once *S. aureus* was identified from a blood culture, a microbiology laboratory technologist contacted an ASP physician or clinical pharmacist during routine working hours (0800 to 1600, Monday to Friday).
- Added following to Methods Patients were excluded in the intervention group if samples were identified by MALDI-ToF when ASP was not available, primarily outside routine working hours (0800 to 1600, Monday to Friday).

Line 259: as with earlier comment, please look through the chart and describe what confounders are responsible for a difference in hospital length of stay

- Updated

Line 289: typo for CoNS

- Fixed

Table 2: please include mortality in each group

- Reply: Also added re-admission and relapse bacteremia to Table 2.

Supplemental table 2: font size and type is not uniform in this table

- Reply: Fixed.

January 26, 2023

Dr. Paul Bonnar
Nova Scotia Health Authority
5078 Dickson Bldg, 5780 University Ave
Halifax, Nova Scotia B3H2Y1
Canada

Re: Spectrum01648-22R2 (Optimizing treatment of *Staphylococcus aureus* bloodstream infections following rapid molecular diagnostic testing and an antimicrobial stewardship program intervention)

Dear Dr. Paul Bonnar:

Your manuscript has been accepted, and I am forwarding it to the ASM Journals Department for publication. You will be notified when your proofs are ready to be viewed.

Sincerely,

Tulip Jhaveri
Editor, Microbiology Spectrum
